# Machine Learning-Assisted Synchronous Fluorescence Sensing Approach for Rapid and Simultaneous Quantification of Thiabendazole and Fuberidazole in Red Wine

**DOI:** 10.3390/s22249979

**Published:** 2022-12-18

**Authors:** Jia-Rong He, Jia-Wen Wei, Shi-Yi Chen, Na Li, Xiu-Di Zhong, Yao-Qun Li

**Affiliations:** The MOE Key Laboratory of Spectrochemical Analysis & Instrumentation, Department of Chemistry, College of Chemistry and Chemical Engineering, Xiamen University, 422 Siming South Road, Siming District, Xiamen 361005, China

**Keywords:** machine learning, synchronous fluorescence, red wine, pesticide residues, thiabendazole, fuberidazole

## Abstract

Rapid analysis of components in complex matrices has always been a major challenge in constructing sensing methods, especially concerning time and cost. The detection of pesticide residues is an important task in food safety monitoring, which needs efficient methods. Here, we constructed a machine learning-assisted synchronous fluorescence sensing approach for the rapid and simultaneous quantitative detection of two important benzimidazole pesticides, thiabendazole (TBZ) and fuberidazole (FBZ), in red wine. First, fluorescence spectra data were collected using a second derivative constant-energy synchronous fluorescence sensor. Next, we established a prediction model through the machine learning approach. With this approach, the recovery rate of TBZ and FBZ detection of pesticide residues in red wine was 101% ± 5% and 101% ± 15%, respectively, without resorting complicated pretreatment procedures. This work provides a new way for the combination of machine learning and fluorescence techniques to solve the complexity in multi-component analysis in practical applications.

## 1. Introduction

Pesticides have a crucial function in avoiding pests and illnesses and enhancing food production, thus they are widely employed in agricultural production across the world. Nonetheless, pesticide residues not only have an impact on the environment but also on the health of humans [1]. According to studies, pesticides have a long residual life and may readily infiltrate food during processing, which has hazardous effects on humans [2,3]. Therefore, detection and management of pesticide residues, particularly the detection of pesticide content in food, are of vital relevance [4]. Red wine, being one of the most popular beverages in the world, has medical benefits such as the prevention of cardiovascular disease, as well as beauty effect and weight reduction benefits [5,6]. However, residual benzimidazole pesticides from the grape growing process may enter the wine during the winemaking process, causing harm to the drinker [7,8]. Thiabendazole (TBZ) and fuberidazole (FBZ), two benzimidazole pesticides, are particularly widely used in grape growth. Various countries attach great importance to benzimidazole pesticide residues in food and have established maximum residue limits ranging from 0.01 to 10 mg/kg, respectively [9]. Moreover, according to the US Environmental Protection Agency (EPA) and World Health Organization (WHO), the allowable daily intake of TBZ is 0.1 and 0.3 mg/kg [10]. Thus, our research on the rapid detection of TBZ and FBZ in red wine has the significance of protecting the life and health of consumers.

Conventional approaches for detecting benzimidazole pesticides include liquid chromatography-mass spectrometry and high-performance liquid chromatography [11,12,13,14]. However, the current approaches are typically expensive and time consuming, limiting wide application. Li. et al. [15] has quantified TBZ and FBZ in red wine using a combination of three-dimensional fluorescence spectroscopy and trilinear decomposition techniques, but this method is still time consuming because it is based on the acquirement of full fluorescence matrix data. Thus, it is necessary to develop a simple, fast, and inexpensive technique for detecting benzimidazole pesticides.

Synchronous fluorescence spectrometry, a technique initially described by Lloyd [16] in 1971 to map fluorescence spectra in the presence of simultaneous scanning of both excitation and emission monochromators, has been widely utilized for the investigation of mixtures. The simultaneous fluorescence approach is used by a growing number of scientists for the determination of compounds in complicated matrices, and it may be combined with other techniques, such as derivative technique, to accomplish further spectral separations [17,18,19,20,21,22]. Among these, the constant-energy synchronous fluorescence approach utilized in this work can effectively simplify the spectra, narrow the spectral band, and eliminate the influence of scattered light by selecting an appropriate constant-energy difference, making it advantageous for the detection of mixtures.

In recent years, with the continuous development of machine learning technology, its combined use with spectroscopy has also increased, allowing for the separation and identification of mixtures by machine learning combined with spectroscopic methods [23,24,25,26,27,28,29,30,31,32,33,34]. The mathematical separation method provided by the machine learning method opens up a new way for spectral analysis; its combination with synchronous fluorescence spectrometry can realize rapid detection of substances with serious overlap.

In this study, we investigated the effect of the second-order derivative constant-energy simultaneous fluorescence sensing technique on the separation of TBZ and FBZ and established the corresponding dataset. We then examined the performance of different machine learning methods on our dataset, built the appropriate models, and tested the models’ effectiveness on independent test sets. Compared with the traditional detection methods, this paper proposes a novel detection method to achieve rapid detection of TBZ and FBZ in red wine.

## 2. Materials and Methods

### 2.1. Reagent and Materials

The standards for TBZ and FBZ were purchased from Beijing Bailingway Technology Co., Ltd., Beijing, China. Amounts of 100 mg/L TBZ-ethanol stock solution and 100 mg/L FBZ-ethanol stock solution were prepared, respectively. The standard stock solution was diluted to the necessary concentration with ethanol to create the standard working solution, and all solutions were stored at 4 °C in a light-resistant environment. The red wine samples were commercially available wines and were kept in a refrigerator at 4 °C in darkness.

### 2.2. Instruments

The scanning of the fluorescence spectra was performed using a laboratory-constructed multifunctional fluorescence spectrophotometer [15,35,36]. The instrument was equipped with a xenon lamp light source of 150 W, slit passband of 5 nm for the excitation and emission monochromators, negative high voltage of −700 V, a 1 cm × 1 cm quartz cuvette for spectral detection, and software to control the fluorimeter written in Turbo C 2.0. Ultrasonic extractors (250 W, 59 kHz) were manufactured by Ningbo Xinzhi Biotechnology Co., Ltd., Ningbo, China. Rotary evaporators (RE-52C) were manufactured by Gongyi Yuhua Instruments Co., Ltd., Gongyi, China. The design of the mathematical model was completed by Matlab R2021a.

### 2.3. Methods

#### 2.3.1. Pre-Treatment Method of Wine Samples

In this study, two pre-treatment methods were evaluated.

Extraction method: An amount of 1.0 mL red wine sample was placed in a 100 mL conical flask and 2 mL ultrapure water was added. The solution was then ultrasonic extracted with 5 mL dichloromethane for 5 min, and the dichloromethane layer was collected. The extraction process of the red wine layer was repeated three times, and the combined dichloromethane layer was collected and spun dry with the rotary evaporator. The product was dissolved in ethanol to fix the volume of 5 mL for measurement.

Dilution method: the red wine was diluted 50 times with ethanol and spiked with the standard as the sample to be examined.

#### 2.3.2. Calibration Set and Test Set

The calibration set and the test set are the datasets from which the computer learns to produce models and, correspondingly, to validate the correctness of the generated models. In this work, a series of samples containing a mixture of TBZ and FBZ were created and randomly separated into calibration and test sets. The calibration set and test set in this study are described in detail in Result and Discussion.

#### 2.3.3. Detection Method

All fluorescence spectra were recorded by a fluorescence sensor at a scan rate of 240 nm/min, and the second derivative constant-energy synchronous fluorescence spectra were recorded by an electronic differential system attached to the spectrofluorometer.

Several models, including a linear regression model, a Gaussian regression model, a support vector regression model, a decision tree model, and an artificial neural network, were tested on the spectral data, and the hyperparameters were optimized by Bayesian search, random search, and grid search in order to establish a suitable model for simultaneous quantitative analysis of TBZ and FBZ in red wine. In this study, the ten-fold cross-validation method is used to improve the generalization performance of the model, the learning algorithm adapted to the data features is selected, and the hyperparameters are optimized for the algorithm. A suitable model is developed, and the model is applied to test actual samples to validate the model’s applicability.

## 3. Result and Discussion

### 3.1. Fluorescence Spectral Analysis and Solvent Selection

Figure 1 depicts the fluorescence spectra of TBZ and FBZ in water and ethanol. In our investigation, ethanol was chosen as the solvent for the fabrication of standard samples and the dilution of wine samples, because its fluorescence intensity was much larger than that with water.

As shown in Figure 1, the spectra of TBZ and FBZ have substantial overlap, and it is difficult to quantify the mixture simultaneously using conventional fluorescence spectroscopic techniques. Because the physical pre-separation required by conventional fluorescence methods is time consuming and labor intensive, we considered combining second derivative constant-energy synchronous fluorescence spectrometry with machine learning techniques for the simultaneous detection of TBZ and FBZ in red wine.

### 3.2. Extraction Method

#### 3.2.1. Selection of Constant-Energy Difference

During the simultaneous scanning process, the constant-energy synchronous fluorescence method maintains a constant-energy difference, as in Equation (1), between the excitation and emission wavelengths. This method can narrow the spectral band for a better separation of the mixture.(1)$$\Delta v=(\frac{1}{\lambda_{\mathrm{ex}}}-\frac{1}{\lambda_{\mathrm{em}}})\times 10^{7}$$ where the unit of Δ*v* is cm^−1^ and the units of *λ_ex_* and *λ_em_* are nm.

Improving the spectral separation efficiency of this approach hinges on the selection of a proper constant-energy difference. In this work, FBZ (30 ng/mL) and TBZ (300 ng/mL) were chosen as the test samples in conjunction with second derivative constant-energy synchronous fluorescence spectroscopy. In order to make the spectral difference caused by the change of constant energy difference more intuitive, we adjusted the concentration levels of the two substances according to their fluorescence properties, so that they had similar fluorescence intensity.

As shown in Figure 2, the negative peak at 320~340 nm represents the primary distinction between the spectra of TBZ and FBZ. From the figure, we can see that the peak width of the second derivative constant energy synchronous fluorescence spectrum is narrow and the shape of the peak is sharp, which provides convenience for further detection. The selected constant-energy differences have a certain degree of spectral resolution, and for our experiment, we chose the constant-energy difference of 2000 cm^−1^. 

Nonetheless, the figure demonstrates that the second derivative constant-energy synchronous fluorescence spectra of both TBZ and FBZ at this constant-energy difference cannot be completely separated, so we considered the introduction of machine learning technique for additional processing in order to separate the spectra mathematically.

#### 3.2.2. Calibration Set

The 24 calibration sets consisted of manually generated samples with the concentration composition shown in Table 1, and the resultant second derivative constant-energy synchronous fluorescence spectra are depicted in Figure 3. Concentration ratios were changed to check the mutual interference of TBZ and FBZ to the spectra.

#### 3.2.3. Machine Learning

In this study, a series of regression methods, including linear regression, Gaussian regression, support vector regression, decision tree, and neural network, were investigated, and the hyperparameters were improved using Bayesian search, random search, grid search, etc. The establishment and optimization of the model are mainly completed through the regression learner and regression network of Matlab. A manually formulated test set was used to verify the generalization performance of the model, and the root mean square error (RMSE), as in Equation (2), was established to characterize ve the models: (2)$$\mathrm{RMSE}=\sqrt{\frac{{\left(y_{fit}-y_{0}\right)}^{2}}{n}}$$ where *y_fit_* is the predicted value of the model for calibration set, *y*_0_ is the actual value, and n is the number of samples. 

From Table 2, we can see that the support vector regression model has the best performance. Therefore, we further optimize the regression method to establish a prediction model.

Finally, we developed a support vector regression (SVR) model, which is a model for regression quantification based on a support vector machine model. When given a training sample, as in Equation (3), a model such as Equation (4) is constructed so that *f*(*x*) is as close to y as possible:(3)$$D=\left[\left(x_{1},y_{1}\right),\left(x_{2},y_{2}\right),\ldots ,\left(x_{m},y_{m}\right)\right]\;\mathrm{with}\;y_{i}\in R$$
(4)$$f\left(x\right)=w^{T}x+b$$where *x* and *b* are the model parameters to be confirmed.

With the derivative constant energy simultaneous fluorescence sensing technique, we obtained the wavelength points of the corresponding calibration set. We took these fluorescence data as inputs and the concentrations of the corresponding TBZ and FBZ as outputs. Through ten-fold cross-validation, we adjusted the hyperparameters and obtained the best model. Finally, we used the obtained model for the prediction of the test set. The process is shown in Figure 4.

The performance of the model on the 12 test sets is shown in Table 3, with the recovery rate in the range of 85~110%, which meets the requirement of practical testing.

#### 3.2.4. Predicted Results for Actual Samples

The established support vector regression model-assisted ultrasonic extraction method was employed to analyze the adulterated commercially available wines. From Table 4, we can see that the recoveries of TBZ and FBZ are basically within the range of 100% ± 10%. This demonstrates that the established approach can be used for the determination of actual samples.

### 3.3. Dilution Method

In order to simplify and increase the efficiency of the test, we considered diluting the original wine directly and prepared the calibration series and the test series using the same method.

#### 3.3.1. Fluorescence Spectra of Diluted Wine Samples Spiked with TBZ and FBZ

As shown in Figure 5, the excitation and emission spectra of 300 ng/mL TBZ and 30 ng/mL FBZ were examined in order to determine the influence of the wine matrix on the fluorescence characteristics of TBZ and FBZ. From the figure, we can see that there is a difference between the spectra obtained by the dilution method compared with that of the standard sample due to the influence of the red wine matrix. However, for this study, this influence can be learned and compensated for by a machine learning model to establish a fluorescence detection method that combines the dilution and machine learning methods.

#### 3.3.2. Selection of Constant-Energy Difference

For the study of constant-energy difference, a mixed sample of TBZ (300 ng/mL) and FBZ (30 ng/mL) was added to the diluted wine, the constant-energy difference was chosen as shown in Figure 6, and the test results are depicted in the figure. We found that the selected constant energy difference has the separation effect for two pesticides in diluted red wine. For our experiment, a constant-energy difference of 5000 cm^−1^ was chosen. However, we discovered that a constant-energy difference of 5000 cm^−1^ is insufficient to distinguish between the spectra of the two medications with 300~330 nm; thus, it is important to integrate the machine learning approach for identification.

#### 3.3.3. Correction Set

In the dilution method, the calibration set was constructed using diluted wine samples as the matrix. The concentration is shown in Table 5, and the second derivative constant-energy synchronous fluorescence spectra is depicted in Figure 7.

#### 3.3.4. Machine Learning

The selection and optimization process of the machine learning method was identical to that of the extraction method, and the prediction model was established by applying the support vector regression model. The performance of different algorithms is shown in Table 6.

Three kinds of wine samples were used to test the model, and the results are presented in Table 7. By using the dilution method, we find that the recovery rate satisfies the detection requirements, and it can basically achieve the same accuracy as the extraction method.

## 4. Conclusions

In this study, a support vector regression-assisted second derivative constant-energy synchronous fluorescence spectrometric approach was developed to achieve the simultaneous and rapid detection of two benzimidazole pesticides, TBZ and FBZ, in red wine. Either simple extraction or direct dilution is enough for sample pre-treatment, without resorting to complicated procedures of separation and purification. It is an efficient new technique for detecting mixtures in complicated systems.

From the results, the following conclusions can be drawn:The application of a derivative constant-energy synchronous fluorescence sensor preliminarily separates the spectra of the mixture, which facilitates the application of machine learning;Machine learning can further assist the fluorescence sensor to analyze and realize rapid detection of mixtures in a complex matrix;The method proposed in this study is simple and quick, which overcomes the disadvantages of traditional detection methods in terms of being time consuming and having high cost.

## Figures and Tables

**Figure 1 sensors-22-09979-f001:**
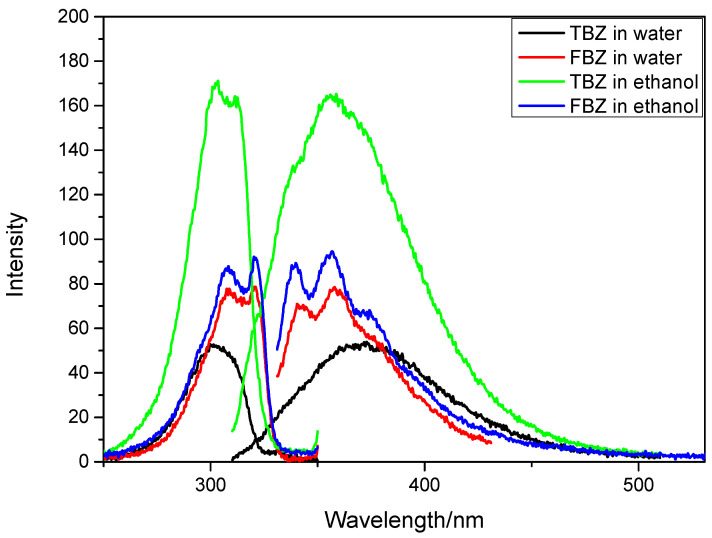
Excitation and emission fluorescence spectra of thiabendazole (TBZ) (300 ng/mL) and fuberidazole (FBZ) (30 ng/mL) in water and ethanol.

**Figure 2 sensors-22-09979-f002:**
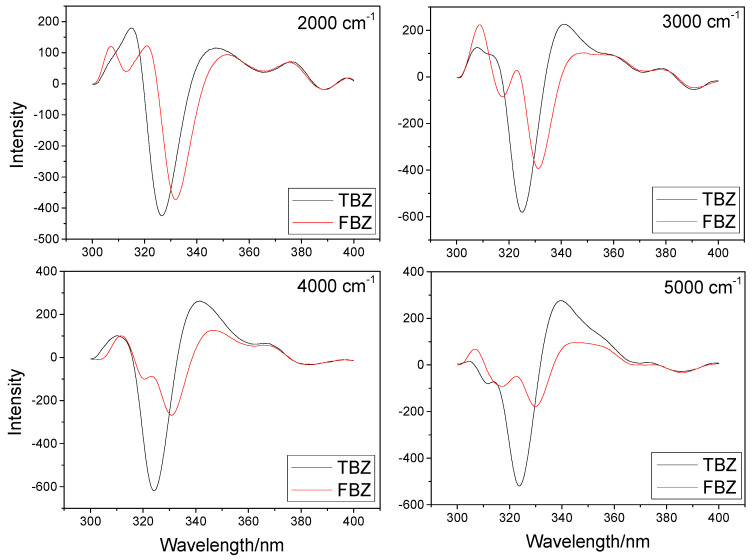
Second derivative constant-energy synchronous fluorescence spectra of TBZ (300 ng/mL) and FBZ (30 ng/mL) at different constant-energy differences (cm^−1^).

**Figure 3 sensors-22-09979-f003:**
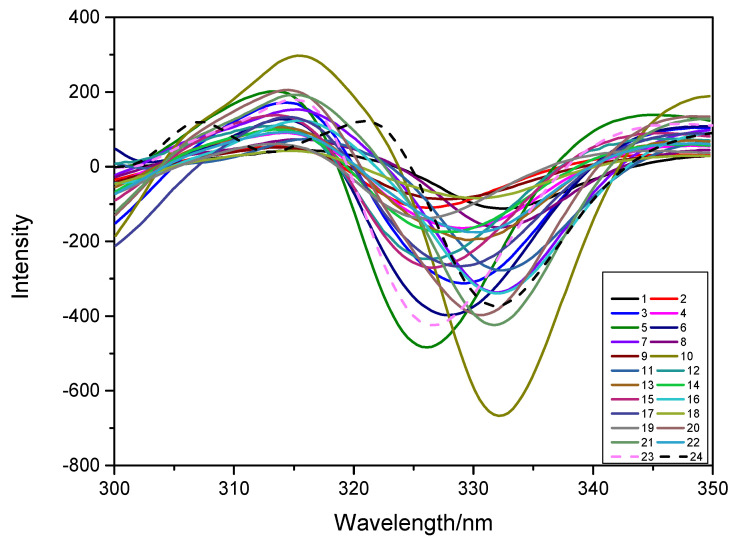
Second derivative constant-energy synchronous fluorescence spectra of 24 calibration sets.

**Figure 4 sensors-22-09979-f004:**
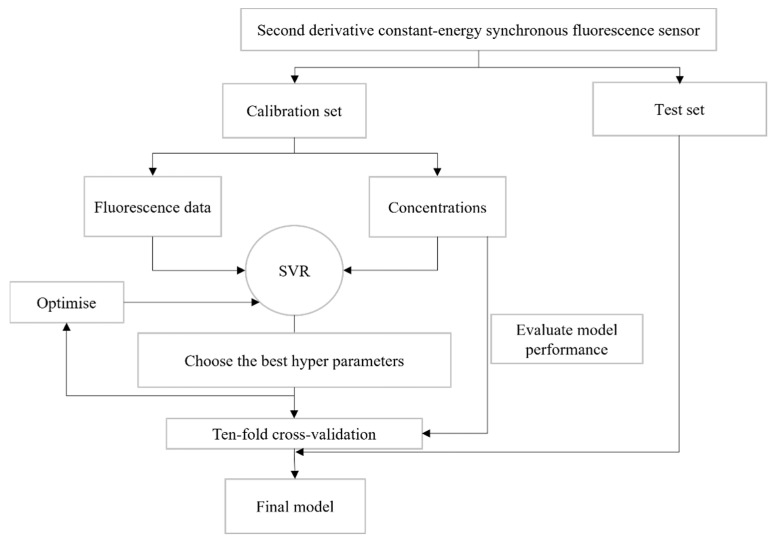
The process of our model building.

**Figure 5 sensors-22-09979-f005:**
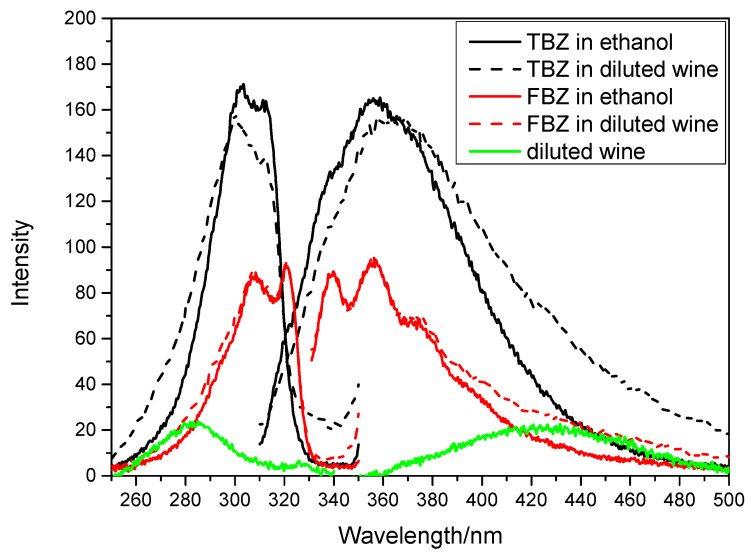
Excitation and emission fluorescence spectra of TBZ (300 ng/mL) and FBZ (30 ng/mL) in ethanol and diluted wine samples and diluted wine.

**Figure 6 sensors-22-09979-f006:**
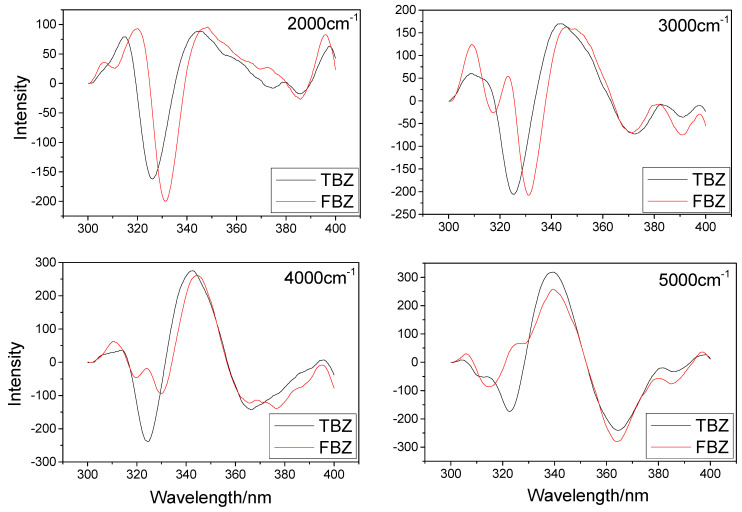
Dilution method: second derivative constant-energy synchronous fluorescence spectra of TBZ (300 ng/mL) and FBZ (30 ng/mL) spiked in diluted wine sample at different constant-energy differences (cm^−1^).

**Figure 7 sensors-22-09979-f007:**
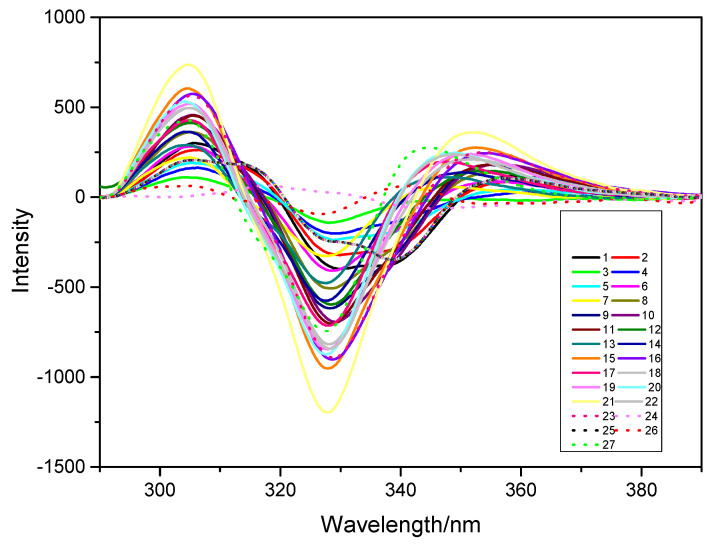
Second derivative constant-energy synchronous fluorescence spectra with 27 calibration sets.

**Table 1 sensors-22-09979-t001:** Concentration of 24 calibration sets.

No.	TBZ (ng/mL)	FBZ (ng/mL)	No.	TBZ (ng/mL)	FBZ (ng/mL)
1	300	40	13	100	20
2	300	50	14	300	30
3	100	50	15	100	40
4	200	30	16	300	10
5	100	10	17	400	50
6	500	20	18	300	20
7	200	10	19	400	30
8	500	10	20	200	20
9	400	40	21	200	50
10	100	30	22	500	30
11	200	40	23	300	0
12	400	10	24	0	30

**Table 2 sensors-22-09979-t002:** Performance of Different Algorithms for Extraction Method.

Regression Methods	RMSE for TBZ	RMSE for FBZ
linear regression	57.29	8.41
Gaussian regression	57.56	7.02
support vector regression	45.19	4.86
decision tree	76.84	9.25
neural network	67.13	11.03

**Table 3 sensors-22-09979-t003:** Predicted results of the developed prediction model for the manually formulated test set.

Actual		Predict		Recovery	
TBZ (ng/mL)	FBZ (ng/mL)	TBZ (ng/mL)	FBZ (ng/mL)	TBZ (%)	FBZ (%)
250	15	264	16.5	105.8	109.9
800	20	705	18.5	88.2	92.6
150	10	159	8.5	105.8	85.2
250	20	256	18.9	102.3	94.2
300	0	295	0	98.3	/
200	50	190	51.5	95.2	103.0
500	10	484	10.4	96.7	104.3
600	20	642	21.5	107.0	107.4
300	30	283	31.5	94.3	104.9
500	0	490	0	98.1	/
500	25	511	26	102.1	104.1
0	25	0	23.5	/	94.1
**Average recovery**				99 ± 10	100 ± 15

**Table 4 sensors-22-09979-t004:** Predicted results of the actual spiked samples.

Actual		Predict		Recovery	
TBZ (ng/mL)	FBZ (ng/mL)	TBZ (ng/mL)	FBZ (ng/mL)	TBZ (%)	FBZ (%)
200	70	230.2	72.2	115.1	103.1
150	60	158.9	56	105.9	93.4
100	50	91.5	49.3	91.5	98.7
**Average recovery**				104 ± 10	98 ± 5

**Table 5 sensors-22-09979-t005:** Concentration of 27 calibration sets.

No.	TBZ (ng/mL)	FBZ (ng/mL)	No.	TBZ (ng/mL)	FBZ (ng/mL)
1	100	50	15	400	50
2	100	40	16	400	40
3	100	10	17	400	10
4	100	20	18	400	30
5	100	30	19	500	30
6	200	30	20	500	20
7	200	10	21	500	50
8	200	40	22	500	10
9	200	50	23	500	40
10	300	50	24	0	10
11	300	40	25	0	50
12	300	30	26	100	0
13	300	10	27	500	0
14	300	20			

**Table 6 sensors-22-09979-t006:** Performance of Different Algorithms for Extraction Method (Machine Learning).

Regression Methods	RMSE for TBZ	RMSE for FBZ
linear regression	71.07	7.39
Gaussian regression	43.15	3.78
support vector regression	31.07	2.77
decision tree	75.52	8.35
neural network	108.38	9.20

**Table 7 sensors-22-09979-t007:** Predicted results of dilution method combined with support vector regression model.

Wine Brands	Actual		Predict		Recovery	
	TBZ (ng/mL)	FBZ (ng/mL)	TBZ (ng/mL)	FBZ (ng/mL)	TBZ (%)	FBZ (%)
1	200	0	212	0	106.2	/
	0	30	0	27	/	92.6
	200	20	187	21	93.8	107.3
	400	20	384	21	96.0	105.2
	150	10	137	8	91.8	85.7
	250	50	231	52	92.6	104.5
	300	0	318	0	106.2	/
	400	35	418	37	104.6	106.4
	500	15	481	12	96.3	85.2
	500	25	495	22	99.0	91.1
2	400	0	418	0	104.6	/
	0	40	0	42	/	105.6
	200	20	217	21	108.7	108.2
	150	20	168	19	112.3	98.6
	250	10	268	12	107.4	122.2
3	400	20	401	18	100.4	93.8
	250	0	231	0	92.6	/
	500	20	518	22	103.7	111.1
**Average recovery**					101 ± 5	101 ± 15

## Data Availability

The data presented in this study are available on request from the corresponding author.

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
