# Peer review of "Machine Learning-Assisted Synchronous Fluorescence Sensing Approach for Rapid and Simultaneous Quantification of Thiabendazole and Fuberidazole in Red Wine"

_sensors, 2022, doi:10.3390/s22249979_

Round 1

Reviewer 1 Report

The novelty of the manuscript is moderated. Synchronous fluorescence has been already used for classification of wines:

-       https://doi.org/10.1007%2Fs13197-020-04291-y;

-       Andreeva, Y., Borisova, E., Genova, T., Zhelyazkova, A., & Avramov, L. (2015, January). Synchronous fluorescence spectroscopy for analysis of wine and wine distillates. In 18th International School on Quantum Electronics: Laser Physics and Applications (Vol. 9447, pp. 264-269). SPIE.

Or even for the determination of TBZ: 

-       Zhong, X. D., Fu, D. S., Wu, P. P., Liu, Q., Lin, G. C., Cao, S. H., & Li, Y. Q. (2014). Rapid fluorescence spectroscopic screening method for the sensitive detection of thiabendazole in red wine. Analytical Methods6(18), 7260-7267)

The only novelty is the simultaneous determination of TBZ and FBZ by machine learning but the authors have already published a paper on the simultaneous determination of both species:

-       Three-Dimensional Fluorescence Technique Coupled with Chemometric second-Order Calibration Method for Simultaneous Detection of Thiabendazole and Fuberidazole in Red Wine. Life Science Instruments 2015, 13 (03), 4.

Apart from that:

-       Is the method valid for the determination of both compounds in other actual samples? This should be discussed.

-       The manuscript lacks interference study. The response to other fluorescent species (not only pesticides) which can be usually found in wine samples should be studied.

-       Justify the concentrations used in the calibration study and why the concentration used in the Selection of Constant-energy Difference is different for each compound.

-       Omit rapid: the optimization and calibration sets need a high number of experiments.

-       Fig. 1 only shows the fluorescence spectra of the compounds in TWO solvents (it is not appropriate to write “various”)

Author Response

We would like to express our appreciations for your review. Your kindly suggestions and comments are very helpful. Please see the attachment for our reponse.

Reviewer 2 Report

The paper “Machine Learning-Assisted Synchronous Fluorescence Sensing Approach for Rapid and Simultaneous Quantification of Thiabendazole and Fuberidazole in Red Wine” reports the results of detection of TBZ and FBZ pesticides in wine using the synchronous fluorescence together with Support vector machine (SVM) and support vector regression (SVR). The paper doesn´t shows new contribution and the results showed in the actual form is not clear and didn´t support adequately the conclusion. The paper needs to correct the following points:

1.- In the introduction, the paper needs to report the safe concentration of TBZ and  FBZ for human life according to World Health Organization (WHO) and why the present work contribute with this condition.

2.- In the introduction the author suggest that the synchronous fluorescence technique eliminate the scattered light. They need to explain why scattering process is talking about, since the actual benchtop spectrometer using laser excitation doesn´t shows this problem. The Fluorescence equipment that use the double monochromators (excitation detection) shows Bragg scattering effect.

3.- The pre-treatment method of wine sample description is very confusing, for example what the meaning “1.0 mL plus standard sample was removed”.

4.- In the section of “calibration set and test set” the authors need to explain who are the calibration and test samples, how many samples of each set were used in this work. What was the criterion used to define the characteristics of the calibration and test samples.

5.- In the section of “Detection Methods” the authors claimed that they used several models, linear regression model, Gaussian regression model, support vector regression, decision tree model, and artificial network. However, any results of all these methods were showed in this work, only was mentioned the results of support vector regression.

6.- In the equation (1) the authors need to indicate the units of Dg, λ and why the factor 107 is used.

7.- Why the authors didn´t shows the synchronous fluorescence spectra curves since these results are the most important issue of this work, since Figures 1 and 4 showed the conventional Fluorescence spectra.

8.- The second derivative of synchronous fluorescence spectra showed smooth curve, why these are possible since the Florescence spectra showed in Figs 1 and 4 are noise curves.

9.- In the section 3.2.3 “Machine learning”. The authors need to describe in the clear way why to use for classification of TBZ and FBZ moieties from the experimental results of the supposed mixture sample and what the support vector regression process was used for. What were the inputs and outputs of each process classification and regression respectively. Since the authors suggest that this article showed contribution in these points, they need to explain in the clear way the model used in the classification and regression process. Since the text in the actual state didn’t give any information about those points above mentioned.

10. – In section 3.3.2 “Selection of constant-energy Difference” The authors were chosen a 5000 cm-1, however the Figure 5 showed that this case is the worst condition so why to choose this condition.

Author Response

(The authors gave the same response as above.)

Round 2

Reviewer 1 Report

Authors have answered most of the queries of the first report.

Reviewer 2 Report

The authors asnwered all the questions suscesfully